# TRAINING NORMALIZING FLOWS FROM DEPENDENT DATA

## ABSTRACT

Normalizing flows are powerful non-parametric statistical models that function as a hybrid between density estimators and generative models. Current learning algorithms for normalizing flows assume that data points are sampled independently, an assumption that is frequently violated in practice, which may lead to erroneous density estimation and data generation. We propose a likelihood objective of normalizing flows incorporating dependencies between the data points, for which we derive a flexible and efficient learning algorithm suitable for different dependency structures. We show that respecting dependencies between observations can improve empirical results on both synthetic and real-world data.

## 1 INTRODUCTION

Density estimation and generative modeling of complex distributions are fundamental problems in statistics and machine learning and significant in various application domains. Remarkably, normalizing flows (Rezende & Mohamed, 2015; Papamakarios et al., 2021) can solve both of these tasks at the same time. Furthermore, their neural architecture allows them to capture even very high-dimensional and complex structured data (such as images and time series). In contrast to other deep generative models such as variational autoencoders (VAEs), which only optimize a lower bound on the likelihood objective, normalizing flows optimize the likelihood directly.

Previous work on both generative models and density estimation with deep learning assumes that data points are sampled *independently* from the underlying distribution. However, this modelling assumption is oftentimes heavily violated in practice. Figure 1 illustrates why this can be problematic. A standard normalizing flow trained on dependent data will misinterpret the sampling distortions in the training data as true signal (Figure 1c. Our proposed method, on the other hand, can correct for the data dependencies and reconstruct the original density more faithfully (Figure 1d).

The problem of correlated data is very common and occurs in many applications. Consider the ubiquitous task of image modeling. The Labeled Faces in the Wild (LFW, (Huang et al., 2008)) data set consists of facial images of celebrities, but some individuals in the data set are grossly overrepresented. For example, George W. Bush is depicted on 530 images, while around 70% of the individuals in the data set only appear once. A generative model trained naively on these data will put considerably more probability mass on images similar to George W. Bush, compared to the less represented individuals. Arguably, most downstream tasks, such as image generation and outlier detection, would benefit from a model that is less biased towards these overrepresented individuals.

In the biomedical domain, large cohort studies involve participants that oftentimes are directly related (such as parents and children) or indirectly related (by sharing genetic material due to a shared ancestry)—a phenomenon called population stratification (Cardon & Palmer, 2003). These dependencies between individuals play a major role in the traditional analyses of these data and require sophisticated statistical treatment (Lippert et al., 2011), but current deep-learning based non-parametric models lack the required methodology to do so. This can have considerable negative impact on downstream tasks, as we will show in our experiments.

In finance, accurate density estimation and modeling of assets (e.g., stock market data) is essential for risk management and modern trading strategies. Data points are often heavily correlated with one another, due to time, sector, or other relations. Traditionally, financial analysts often use copulas for the modeling of non-parametric data, which themselves can be interpreted as a simplified version

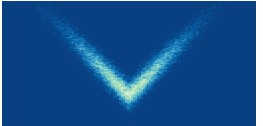 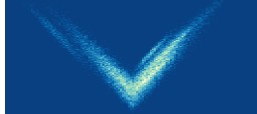 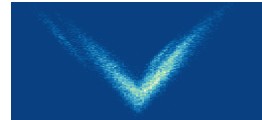 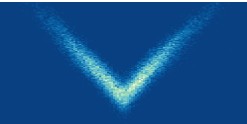

(a) True distribution, sampled independently (b) True distribution, sampled with dependencies (c) Distribution learned by standard normalizing flow, trained on dependent data (d) Distribution learned by normalizing flow adjusted for dependencies, trained on dependent data

Figure 1: Example setting with synthetic data sampled with inter-instance dependencies. Training a standard normalizing flow on this data biases the model. Adjusting for the dependencies during training recovers the true underlying distribution.

of normalizing flows (Papamakarios et al., 2021). Copulas commonly in use, however, are limited in their expressivity, which has led some authors even to blame the 2007-2008 global financial crisis on the use of inadequate copulas (Salmon, 2009). Many more examples appear in other settings, such as data with geospatial dependencies, as well as in time series and video data.

In certain settings from classical parametric statistics, direct modeling of the dependencies in maximum likelihood models is analytically feasible. For linear and generalized linear models, dependencies are usually addressed either with random effects in linear mixed models (Jiang & Nguyen, 2007) or sometimes only by the inclusion of fixed-effects covariates (Price et al., 2006). Recent work in deep learning introduced concepts from random effects linear models into deep learning for *prediction* tasks such as regression and classification (Simchoni & Rosset, 2021; Xiong et al., 2019; Tran et al., 2020). In federated learning of generative models, researchers usually deal with the break of the non-i.i.d. assumptions with ad hoc methods and without consideration of the statistical implications (Augenstein et al., 2020; Rasouli et al., 2020). These methods also only consider block-type, repeat-measurement dependencies for multi-source integration. To the best of our knowledge, both deep generative models and deep density estimation so far lack the tools to address violations against the independence assumption in the general setting and in a well-founded statistical framework.

In this work we show that the likelihood objective of normalizing flows naturally allows for the explicit incorporation of data dependencies. We investigate several modes of modeling the dependency between data points, appropriate in different settings. We also propose efficient optimization procedures for this objective. We then apply our proposed method to three high-impact real-world settings. First, we model a set of complex biomedical phenotypes and show that adjusting for the genetic relatedness of individuals leads to a considerable increase in statistical testing power in a downstream genome-wide association analysis. Next, we consider two image data sets, one with facial images, the other from the biomedical domain, leading to less biased generative image models. In the last application, we use normalizing flows to better model the return correlations between financial assets. In all experiments, we find that adjustment for dependencies can significantly improve the model fit of normalizing flows.

## 2 METHODS

In this section we describe our methodology for training normalizing flows from dependent data. First, we will derive a general formulation of the likelihood under weak assumptions on the dependencies among observations. Afterwards, we will investigate two common settings in more detail.

### 2.1 BACKGROUND: LIKELIHOOD WITH INDEPENDENT OBSERVATIONS

A normalizing flow is an invertible function $t : \mathbb{R}^p \to \mathbb{R}^p$ that maps a $p$-dimensional noise variable $u$ to a $p$-dimensional synthetic data variable $x$. The noise variable $u$ is usually distributed following a simple distribution (such as a $\mathcal{N}_p(0, I_p)$ or $\mathcal{U}_{[0,1]^p}$), for which the density is explicitly known and efficiently computable. By using the change of variable formula, the log-density can be explicitly computed as

$$\log(p_x(x)) = \log(p_u(u)) - \log(|\det J_t(u)|),$$

where $u := t^{-1}(x)$ and $J_t(u)$ is the Jacobian matrix of $t$ in $u$.

Given a data set $x_1, \ldots, x_n$, if the observations are **independent** and identically distributed, the full log-likelihood function readily factorizes into its respective marginal densities:

$$\log(p_x(x_1, \ldots, x_n)) = \sum_{i=1}^{n} \log(p_x(x_i)) = \sum_{i=1}^{n} \log(p_u(u_i)) - \log(|\det J_t(u_i)|).$$

The function $t$ is usually chosen in such a way that both the inverse $t^{-1}$ and the determinant of the Jacobian $J_t$ can be efficiently evaluated, e.g. using coupling layers (Dinh et al., 2017). Therefore, all of the terms in the likelihood can be explicitly and efficiently computed and the likelihood serves as the direct objective for optimization.

## 2.2 LIKELIHOOD WITH DEPENDENCIES

Assuming the data points are identically distributed, but **not independently** distributed, the joint density does not factorize anymore. A model trained on non-independent data but under independence assumptions will hence yield biased results for both density estimation and data generation.

We can derive the non-independent setting as follows. Let $T : \mathbb{R}^{n \times p} \to \mathbb{R}^{n \times p}$ be the normalizing flow applied on all data points together, i.e.,

$$U = T^{-1}(X) = T^{-1}(x_1, \ldots, x_n) = \begin{pmatrix} t^{-1}(x_1)^\top \\ \ldots \\ t^{-1}(x_n)^\top \end{pmatrix}.$$

$X, U \in \mathbb{R}^{n \times p}$ are now matrix-variate random variables. We can still apply the change of variable formula, but on the $n \times p \to n \times p$ transformation $T$, instead of the $p \to p$ transformation $t$:

$$\log(p_X(X)) = \log(p_U(U)) - \log(|\det J_T(U)|).$$

If $T$ is understood on $np$ instead of $n \times p$ space, it becomes clear that the Jacobian $J_T$ is a block-diagonal matrix,

$$J_T(U) = \begin{pmatrix} J_t(u_1) & 0 & \ldots & 0 \\ 0 & J_t(u_2) & & \\ \ldots & & & \\ 0 & & \ldots & J_t(u_n) \end{pmatrix},$$

for which the determinant is readily available: $\det J_T(U) = \prod_{i=1}^{n} J_t(u_i)$. In other words, the log-abs-det term in the normalizing flow objective remains unchanged even under arbitrary dependence structure.

The density $p_U(U)$, however, is challenging and generally not tractable, and we will consider different assumptions on the joint distribution of $U$.

In the most general case, we could assume that each $u_i$ is marginally distributed as a $\mathcal{U}_{[0,1]^p}$ variable, with arbitrary dependence structure across observations. This is a direct extension of standard copulas to matrix-variate variables. As learning general copulas is extremely challenging even in relatively low dimensional settings (Jaworski et al., 2010), we focus in this work only on the equivalent of a Gaussian copula:

**Assumption 2.1.** We assume that the dependency within $U$ can be modeled by a matrix normal distribution $\mathcal{MN}$ with independent columns (within observations), but correlated rows (between observations):

$$U \sim \mathcal{MN}_{n,p}(0, C, I_p) \triangleq \mathcal{N}_{np}(0, I_p \otimes C).$$

Here, $\otimes$ denotes the Kronecker product.

We can model the columns of $U$ with a 0-mean vector and $I_p$-covarianace, as the normalizing flow $t$ is usually chosen to be expressive enough to transform a $\mathcal{N}_p(0, I_p)$ into the desired data distribution. We note that this assumption means that we cannot model all forms of latent dependencies so it constitutes a trade-off between expressivity and tractability.

Now we can state the full likelihood in the non-i.i.d. setting:

$$\log(p_X(X)) = -\sum_{i=1}^{n} \log(|\det J_t(u_i)|) - \frac{np}{2}\log(2\pi) - \frac{p}{2}\log(\det(C)) - \frac{1}{2}tr(U^\top C^{-1}U). \quad (1)$$

## 2.3 SPECIFIC COVARIANCE STRUCTURES

We investigate different assumptions on the covariance structure in the latent dependency model. The most general case is a fully unspecified covariance matrix, e.g. parametrized as the lower-triangular Cholesky decomposition of its inverse, $C = L^{-1}L^{-\top}$ with $n(n + 1)/2$ parameters. In this case, the determinant can be efficiently computed, as $\det(C) = \det(L^{-1}L^{-\top}) = \det(L^{-1})^2 = \prod_{i=1}^{n}(L^{-1})_{i,i}^2$. Matrix products with $C^{-1}$ can also be evaluated reasonably fast. However, this parametrization requires optimizing $O(n^2)$ additional parameters, which is unlikely to yield useful estimates and may be prone to overfitting.

Instead, we consider two different assumptions on $C$ that are very common in practice and give a reasonable trade-off between expressivity and statistical efficiency.

### 2.3.1 KNOWN AND FIXED COVARIANCE MATRIX

In many settings, side information can yield relationship information, given in the form of a fixed relationship matrix $G$. The covariance matrix then becomes $C = \lambda I_n + (1 - \lambda)G$ with only parameter $\lambda \in [0, 1]$ to be determined.

This setting is commonly assumed for confounding correction in genetic association studies, where $G$ is a genetic relationship matrix (where the entries are pairwise genetic relationships computed from allele frequencies (Lippert et al., 2011)) or based on pedigree information (e.g., a parent-child pair receives a relationship coefficient of 0.5 and a grandparent-grandchild pair of 0.25 (Visscher et al., 2012)). Similarly, for time-related data, we can define relationship via, e.g., a negative exponential function: $C_{i,j} = \exp(-\gamma(t_i - t_j)^2)$, where the hyperparameter $\gamma > 0$ is a time-decay factor and $t_i$ and $t_j$ are the measurement time points of observations $i$ and $j$, respectively.

More generally, $G$ itself can again be a mixture of multiple relationships $G = \sum_{r=1}^{R} G_r$, where $G_r$ denote multiple sources of relatedness. In this work, we consider $G$ to be fully specified and only estimate $\lambda$.

If the sample size is moderate (say, below 50k), an efficient approach to optimizing $\lambda$ (Lippert et al., 2011) consists of first computing the spectral decomposition of $G = Q\Lambda Q^\top$ (with diagonal $\Lambda$ and orthogonal $Q$) and noticing that $\lambda I_n + (1-\lambda)G = Q(\lambda I_n + (1-\lambda)\Lambda)Q^\top$. Then, the log-determinant and the trace are

$$\log(\det(C)) = \sum_{i=1}^{n} \log(\lambda + (1 - \lambda)\Lambda_{i,i}), \text{ and}$$

$$tr(U^\top C^{-1}U) = tr((Q^\top U)^\top (\lambda I_n + (1 - \lambda)\Lambda)^{-1}Q^\top U).$$

The rotation matrix $Q$ makes mini-batch estimation of the trace term inefficient, as $Q$ will either mix $U$ across batches or requires a full re-evaluation of $Q(\lambda I_n + (1 - \lambda)\Lambda)^{-1}Q^\top$ after each update to $\lambda$, i.e., in every mini-batch. Instead, we optimize the parameters of the normalizing flow and $\lambda$ in an alternating two-step procedure, see Section 2.4.2. Note that the main additional cost of this procedure, the spectral decomposition of $G$, is independent of $\lambda$ and only needs to be performed once for a given relationship matrix $G$.

For larger sample sizes, there still exist practical algorithms for estimating the variance component (Loh et al., 2015). In practice, $G$ is also often sparse or can be approximated sparsely (e.g., by setting all elements with absolute value below a fixed threshold to 0). This can greatly accelerate parameter estimation and is usually accurate enough in practice (Jiang et al., 2019). More generally, different matrix structures may allow for additional speed-ups, but we defer this investigation to future work.

### 2.3.2 BLOCK-DIAGONAL, EQUICORRELATED COVARIANCE STRUCTURE

In the next setting, we consider a *block-diagonal* covariance matrix $C$ with *equicorrelated correlation matrices* $C_i \in \mathbb{R}^{n_i \times n_i}$ as blocks:

$$C = \begin{pmatrix} C_1 & 0 & \ldots & 0 \\ 0 & C_2 & & \\ \ldots & & & \\ 0 & \ldots & & C_N \end{pmatrix}, \quad \text{where} \quad C_i = \begin{pmatrix} 1 & \rho_i & \ldots & \rho_i \\ \rho_i & 1 & & \\ \ldots & & & \\ \rho_i & \ldots & & 1 \end{pmatrix}.$$

with $\rho_i \in (0,1)$ (we ignore the case of potentially anti-correlated blocks). In other words, there is no dependence between blocks, and there is a constant dependence within blocks. We assume that the *block structure* is known ahead and we only need to find the parameters $\rho_i$. For each block there is either no ($n_i = 1$) or only one ($n_i > 1$) parameter to be learned.

The assumption of equicorrelated blocks is reasonable in settings with *repeat measurements* of identical objects or individuals. E.g., in a facial image data set, certain individuals may have multiple images. This setting is similar to the setting of high-cardinality categorical features in prediction models (Simchoni & Rosset, 2021).

Both the determinant and the inverse of each block can be efficiently computed ((Tong, 2012), Prop. 5.2.1 & 5.2.3):

$$\det(C_i) = (1 + (n_i - 1)\rho_i)(1 - \rho_i)^{n_i - 1}, \quad \text{and} \quad (C_i^{-1})_{j,k} = \begin{cases} \frac{1 + (n_i - 2)\rho_i}{(1 - \rho_i)(1 + (n_i - 1)\rho_i)} & \text{if } j = k \\ \frac{-\rho_i}{(1 - \rho_i)(1 + (n_i - 1)\rho_i)} & \text{otherwise.} \end{cases}$$

### 2.4 OPTIMIZATION

#### 2.4.1 MINI-BATCH ESTIMATION

The full likelihood in Equation 1 can be computed explicitly but does not lend itself easily to stochastic optimization with mini-batches. Note that the log-abs-det term decomposes nicely into independent observations and the next two terms are independent of the observations. Only the trace term is problematic for mini-batch estimation, so we propose an unbiased stochastic estimator for it.

**Proposition 2.2.** *Given a mini-batch of size $b \geq 2$ and $\xi \in \{0,1\}^n$ a variable indicating batch inclusion (i.e., $x_i$ is in batch iff $\xi_i = 1$; $\sum_{i=1}^n \xi_i = b$) and $A := C^{-1}$, the stochastic trace estimator*

$$\bar{tr}_\xi = \frac{n}{b} \sum_{i=1}^n \xi_i A_{i,i} u_i^\top u_i + 2 \frac{n(n-1)}{b(b-1)} \sum_{i<j} \xi_i \xi_j A_{i,j} u_i^\top u_j \tag{2}$$

*is unbiased, i.e., $\mathbb{E}_\xi[\bar{tr}_\xi] = tr(U^\top A U)$.*

The proof can be found in Appendix A. The trace estimator $\bar{tr}_\xi$ only depends on observations $u_i = t^{-1}(x_i)$ within the batch and can be efficiently computed, assuming $A = C^{-1}$ can be efficiently evaluated, which is the case for the parametrizations discussed in Section 2.3.

#### 2.4.2 TRAINING SCHEDULES

From here on, we distinguish between the true parameters $\lambda$ and $\rho_i$, and the parameters estimated by our model, $\hat{\lambda}$, $\hat{\rho}_i$.

**Known & Fixed Covariance** Joint optimization between $\hat{\lambda}$ and the parameters of the flow is possible, but would require in each step a full re-evaluation $tr(U^\top C^{-1} U)$ across the full data set, instead of just the current mini-batch. This makes this training scheme infeasible. Instead, we propose two different methods to optimize both the flow parameters and variance component $\hat{\lambda}$. First, we can use a simple **grid search** over different possible values for $\hat{\lambda}$ and choose the best according to performance on a validation set.

Second, we can use an **alternating descent** approach. In this case, we alternate between optimizing only the parameters of the flow model for a number of epochs (with a version of mini-batch stochastic gradient descent) and only optimizing $\hat{\lambda}$ for a number of epochs (with gradient descent). At the beginning of every flow-parameter training stage, we compute the current $A = C^{-1}$ for the given $\hat{\lambda}$ and can then compute all mini-batch likelihood estimates using the trace estimator in Equation 2 without the need for recomputation. At the beginning of every $\hat{\lambda}$ training stage, we only once compute the rotated noise variables $Q^\top U$ for the full data set and can then optimize the derivative of the full objective with respect to $\hat{\lambda}$ very efficiently. The trace can be computed as $tr((Q^\top U)^\top (\hat{\lambda} I_n + (1 - \hat{\lambda})\Lambda)^{-1} Q^\top U)$ or as $tr(Q^\top U (Q^\top U)^\top (\hat{\lambda} I_n + (1 - \hat{\lambda})\Lambda)^{-1})$ due to the cyclical trace property, but in our experiments we found that this was not a bottleneck computation.

Table 1: Results in terms of the test-data negative log-likelihoods for synthetic data with equicorrelated blocks (top) and fixed covariance (bottom), averaged over 10 random seeds (lower = better). Significantly better results are in bold (one-sided paired t-test, $\alpha = 0.05$). Baseline is the same model without taking dependencies into consideration.

|  | Algorithm | Abs | Crescent | CrescentCubed | Sign | SineWave |
|---|---|---|---|---|---|---|
| Equicorrelated Blocks | Baseline | 1.513 | 2.021 | 3.010 | 1.519 | 2.070 |
|  | Grid Search | **1.379** | **1.885** | **2.938** | **1.420** | **1.983** |
|  | Joint | 1.475 | 2.005 | 3.067 | 1.501 | 2.087 |
| Fixed Covariance | Baseline | 1.590 | 2.144 | 3.262 | 1.818 | 2.353 |
|  | Grid Search | **1.376** | **1.866** | **2.944** | **1.443** | **2.040** |
|  | Alternating | **1.361** | **1.869** | **2.915** | **1.430** | **2.064** |

To yield values in the interval $[0, 1]$, we chose to parametrize $\hat{\lambda}$ as the output of a sigmoid function $\hat{\lambda} = \sigma(\hat{\lambda}_{raw})$, where $\hat{\lambda}_{raw} \in \mathbb{R}$ is the raw optimization parameter. We tried different sigmoidal parametrizations, but those had little effect on the outcome.

**Equicorrelated Blocks** In the case of equicorrelated blocks, we also propose two different training schemes. First, we can again use a simple **grid search** over a single joint parameter $\hat{\rho} = \hat{\rho}_1 = \ldots = \hat{\rho}_N$. Alternatively, if there are only very few blocks, a grid search for all $\hat{\rho}_i$ is possible, although the exploration space grows exponentially with the number of blocks $N$.

Second, due to the simple computations of $\det(C)$ and $C^{-1}$ in this case, we can also perform a **joint** optimization over the flow parameters and all $\hat{\rho}_i$. We again parametrize $\hat{\rho}_i$s with raw parameters pushed through a sigmoid function as for $\hat{\lambda}$.

## 3 EXPERIMENTAL EVALUATION

We validate on both synthetic and real-world data that our novel training scheme can help alleviate sampling biases when training normalizing flows. On real-world data with non-independent data, the ground-truth dependency structure is usually not known, making the evaluation inherently challenging. Therefore, we first investigate simulated settings where we can explicitly control the dependencies. Our evaluation metric in all settings is the negative log-likelihood (NLL) on a holdout test set. For the imaging experiments, we also report bits per dimension (bpd), a linear transformation of the NLL. Additional details for all experiments can be found in Appendix B

### 3.1 SYNTHETIC DATA EXPERIMENTS

#### 3.1.1 EQUICORRELATED DATA

In the first setting, we simulate a draw with repeat measurements, inducing an equicorrelated dependency structure as described in Section 2.3.2. For each block, we draw one $\rho_i \sim \text{Unif}_{[0.5, 0.99]}$ and define the full covariance matrix as in Section 2.3.2. Using this covariance matrix, we sample from several non-parametric 2d distributions provided by Durkan et al. (2019). An example for the Abs data set can be seen in Figure 1. For modelling the equicorrelated blocks, we choose both a grid search over fixed parameters and joint gradient-based optimization of $\hat{\rho}_i$ and the flow parameters.

Table 1 (top part) shows the result. Surprisingly, while the grid search clearly outperforms the baseline, the joint optimization does not improve upon the model. For the Crescent data set we also computed the distance of learned $\hat{\rho}_i$s to true $\rho_i$s for the best models (each block is counted only once, independent of size). The baseline model had an average MSE of 0.57 (MAE: 0.74), while grid search and joint optimization had MSEs of only 0.023 (MAE: 0.13) and 0.08 (MAE: 0.25), respectively. In an additional experiment, this time only on the Crescent data set, we investigate how sensitive our model is to the strength of dependencies. In the data creation, we only change the sampling of true dependency parameters $\rho_i$ from a $\text{Unif}_I$ distribution, with interval $I \in \{[0, 0.2], [0.2, 0.4], [0.4, 0.6], [0.6, 0.8], [0.8, 1.0]\}$. The results are shown in Figure 2.

Table 2: Results on real-world data, negative log-likelihoods on test data set, averaged over 10 random seeds for UKB & Stock Pair data (lower = better). P-values for one-sided paired t-test against baseline in parentheses. Baseline is same model without taking dependencies into consideration.

| Algorithm | UKB Biomarkers | Stock Pairs | ADNI (bpd) | LFW (bpd) |
|---|---|---|---|---|
| Baseline | 24.50 | -5.69 | 7836.5 (2.760) | 6481.6 (3.044) |
| Grid Search | 24.27 $(p = 0.002)$ | -5.72 $(p = 0.002)$ | 7725.8 (2.721) | 6334.7 (2.975) |
| Joint | | -5.71 $(p = 0.003)$ | 7715.8 (2.718) | 6346.5 (2.980) |
| Alternating | 24.04 $(p = 0.00003)$ | | | |

At each of the five data set settings, a one-sided paired t-test shows that the normalizing flow incorporating dependencies outperforms the baseline ($\alpha = 0.05$). As expected, for small dependencies in the true data, both models perform similarly, but our method is very robust and barely decreases in performance up until the highest range of sampling distortions.

### 3.1.2 KNOWN COVARIANCE

We next simulate the setting of a known covariance matrix between different samples but with unknown variance component $\lambda$. We use the covariance structure $\lambda I + (1 - \lambda)G$ as in the equicorrelated case to generate correlated bivariate standard normal samples that again get non-linearly transformed. In Table 1 (bottom part) we compare the results. Both the simple grid search and the alternating descent approach perform considerably better than the naive baseline algorithm that ignores the dependencies in the data.

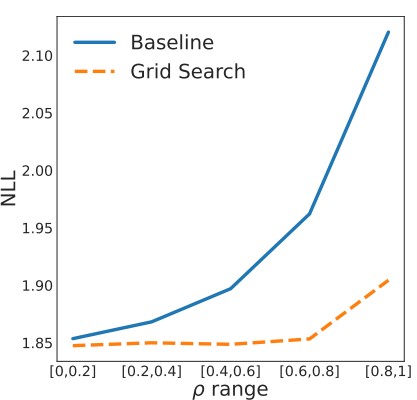

Figure 2: Performance of baseline model versus model adjusted for dependencies, for different strengths of dependencies ($\rho$).

### 3.2 REAL-WORLD DATA

#### 3.2.1 UKB BIOMARKERS

The UK Biobank (UKB, (Bycroft et al., 2018)) provides rich phenotyping and genotyping for a large cross-section of the UK population. We investigate a number of blood biomarkers, whose distribution starkly deviates from standard parametric distribution families. Usually, the data needs to be quantile-transformed to match a normal distribution (Monti et al., 2022), which, however, can decrease power of many statistical methods (McCaw et al., 2020). These biomarkers are well-known to be highly heritable and subject to population stratification, a type of confounding due to joint ancestry of unrelated individuals (Sinnott-Armstrong et al., 2021). We perform two experiments on this data set, building non-parametric density model that can incorporate the distorting genetic correlation between individuals.

**Density modeling** In the first experiment, we select the 3,223 individuals for whom all 30 biomarkers are available. Relatedness between two individuals is computed as the correlation coefficients between the individuals' first 40 (unnormalized) genetic principal components (computed from SNP microarray chip data provided by the UKB resource). We use this Matrix as the fixed covariance structure and optimize over $\hat{\lambda}$. This way of measurement of genetic relatedness between individuals is very common in genetic association studies and has been shown to reliably correct for population stratification (Price et al., 2006). We investigate the density estimation on the test data. Due to the relatively small data set size, we re-run the same experiment 10 times with different random splits between train, validation, and test set and also different network initializations. The results in Table 2, first column, indicate that incorporating the dependencies can significantly improve model fit, both using a grid search and using the alternating optimization scheme.

**Application in association studies** A genome-wide association study (GWAS) is a frequentist hypothesis testing procedure, in which a phenotype is tested for association against a large number of individual genetic mutations (typically on the order of hundreds of thousands or millions of variants). GWAS are a fundamental tool within multiple disciplines in the life sciences, such as in the medical domain, in plant and in lifestock breeding, and have considerably contributed to the understanding of the genetic architecture of complex traits (Visscher et al., 2017). State-of-the-art GWAS algorithms model dependencies between individuals with random effects in a linear mixed model (LMM) framework and can effectively control for both population stratification and (known and cryptic) relatedness between individuals (Yu et al., 2006). In this experiment, we perform *multivariate* GWAS, testing for association between individual genetic variants with multiple phenotypes together.

Due to the high computational cost of multivariate LMMs (mvLMMs), we split the 30 available biomarkers into six disjoint groups of related biomarkers and subsample 10,000 individuals per group. Rank-based normal transformations are insufficient to transform a vector of arbitrarily distributed random variables into a multivariate normal distribution, as would be necessary for mvLMMs. This is due to the fact that not all random vectors whose *marginals* are normally distributed are also multivariate normally distributed; see Figure 3 for an illustration on the biomarker data. Hence, mvLMMs can not be applied to quantile-transformed data. Instead, a standard method is to test for association with each biomarker in the group independently, take the minimum of the p-values over all biomarkers in the group, and perform a Bonferroni-correction for the number of biomarkers in the group. We propose to instead use a normalizing flow to transform the biomarker group into a multivariate normal vector and then apply the mvLMM on this transformed data. We use both a baseline normalizing flow without consideration of the data dependencies, and our proposed method with the *alternating* optimization scheme. More details can be found in Appendix B.2.1.

Table 3: Number of loci associated with biomarker groups at genome-wide significance level, avg. over 3 random seeds. *Single*: univariate, quantile-transformed LMMs; *Baseline*: mvLMM on flow-transformed biomarkers; *Alternating*: mvLMM on biomarkers transformed with flow correcting for dependencies.

| Biomarker group (# biomarkers) | Single | Baseline | Alternating |
|---|---|---|---|
| Bone and joint (4) | 18.7 | 12.0 | 16.3 |
| Cardiovascular (8) | 55.0 | 58.0 | 61.0 |
| Diabetes (2) | 5.3 | 5.3 | 6.7 |
| Hormonal (4) | 6.7 | 5.7 | 7.0 |
| Liver (6) | 29.7 | 32.3 | 35.0 |
| Renal (6) | 18.3 | 19.0 | 18.7 |
| All | 133.7 | 132.3 | 144.7 |

We report the number of indepedent loci significantly associated with each group of biomarkers in Table 3. While the baseline normalizing flow performs similarly to the naive single-dimension approach, taking care of the dependencies can boost the number of found loci by more than 8%.

We believe these findings may also significantly improve the analysis of more complex intermediate phenotypes, such as in full-imaging GWA studies (Kirchler et al., 2022). To the best of our knowledge, this is the first time that normalizing flows have been used for GWAS in this style, although Hansen et al. (2021) recently used normalizing flows in a different GWAS setting.

### 3.2.2 IMAGE MODELING

Image modeling is a major research area for normalizing flows, with applications in image synthesis (Kingma & Dhariwal, 2018), outlier detection (Schirrmeister et al., 2020), and semi-supervised learning (Izmailov et al., 2020). Repeat measurements are very common in image data sets, and without adjusting for dependencies, overrepresentation biases will translate into biased generative models, as well. We investigate two prominent examples.

**ADNI brain imaging** The Alzheimer's Disease Neuroimaging Initiative (ADNI, (Jack Jr et al., 2008)) is a longitudinal study of Alzheimer's Disease (AD) progression, so many of the individuals in the study are imaged multiple times. Prior work on similar data has shown that causal effects can be modeled in generative image models using explicit confounding factors such as age and sex (Pawlowski et al., 2020). Here we show that we can also model the i.i.d.-violations using our proposed method. The data set comprises 1,820 individuals with each individual having between 1 and 35 images (mean: 7.03, median: 6) and a total of 12,799 images. We model these repeat

measurements with the equicorrelated model and use a Glow-type image normalizing flow (Kingma & Dhariwal, 2018) as our base architecture.

**LFW face images** LFW (Huang et al., 2008) consists of 13,233 facial images of 5,749 celebrities, where each individual has between 1 and 530 images (mean: 2.3, median: 1). We again model these repeat measurements with the equicorrelated block model and the same Glow-type architecture as for the ADNI data set.

The results on both data sets show that incorporating dependencies improves the likelihood fit on the holdout test data set. We note that this does not necessarily translate into a higher quality for individual images, but rather into a better fit of the full distribution.

### 3.2.3 STOCK DATA PAIRS

A range of different stock trading and risk management strategies require accurate modeling of the behavior of different stocks (Kole et al., 2007). We focus on modeling the daily returns for two pairs of correlated stocks, which is used, e.g., in pair trading strategies (Stander et al., 2013). A pairs trading strategy can utilize a probabilistic model of stock returns as follows: each day, one can assess if a given stock pair lies outside of a high-confidence region given the model. If the pair behaves anomalously and one stock underperforms compared to the other stock, a trader can hedge these two stocks against each other. The trader would "buy long" the underperforming stock and "sell short" the overperforming stock, with the implicit assumption that in the future the two prices will revert back to a high-confidence region. Here, we use the pairs `AAPL-MSFT` (Apple & Microsoft) and `MA-V` (Mastercard & Visa), each starting from initial public offering (IPO) of the later of the pair, until late 2017, using publicly available data at close time. A single data point is the 2d daily logarithmic return of one of the two pairs of stocks. For example, `MA` closed on 2012/06/21 with a price of \$40.737 and on 2012/06/22 at \$42.080, while `V` closed at \$28.661 and \$29.976 for those two days. The associated data point then is $(\log(42.08/40.737), \log(29.976/28.661)) = (0.0324, 0.0449)$, and a corresponding data point for the same days for `AAPL-MSFT` is in the data set. We split data into train (70%), validation (15%), and test (15%) data temporally (non-randomly) to counteract information leakage. Since Apple and Microsoft had their respective IPOs in the 1980s and Visa and Mastercard theirs in the 2000s, the `AAPL-MSFT` pair is overrepresented in the training data, while both pairs are equally represented in the validation and test data. We use the equicorrelated dependency model with two blocks, one for `AAPL-MSFT` and one for `MA-V`. The distribution fit for the equicorrelated model is slightly improved using the data dependencies, but again shows that a joint optimization appears to be inferior to a simple grid search.

## 4 CONCLUSION

We have shown that through a simple adaptation in the likelihood loss of normalizing flows, we can integrate flexible data dependencies into the training objective, which can also be trained with mini-batch SGD. Experimental evaluation of synthetic and real-world data showed that our method can significantly improve the fit of probabilistic models. In future work, we're especially interested how our method can be extended to other generative models such as VAEs and if it can be combined with other debiasing methods such as causal DAGs as done by Pawlowski et al. (2020).

Our method is not without limitations. The trace estimator in Equation 2 is unbiased, but has a high variance due to the overweighting of the off-diagonal terms. This can lead to unstable gradient estimates, especially in the early stages of training. In addition, joint optimization of $\hat{\rho}_i$s with the flow parameters counterintuitively only sometimes leads to better results. We believe further improvements to the optimization schemes might alleviate these issues.

We also note that, if the goal is density estimation or generative modeling, incorporating dependencies into the normalizing flow objective is not necessary in *all* cases with dependent data. For example, in the case of equicorrelated repeat measurements with *identical block-sizes*, no improvements can be expected. This is because no region of the sampling space is overrepresented relative to the other regions. Only when some blocks are larger than others (or with more general, unbalanced covariance matrices), adjustment for dependencies makes sense. However, if we are interested in *full likelihood evaluation* over the whole data set instead of just density estimation in individual data points, the results will differ.

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

## A  PROOF OF PROPOSITION 2.2

We have

$$\mathbb{E}_\xi[\bar{tr}_\xi] = \frac{n}{b} \sum_{i=1}^n \mathbb{E}_\xi[\xi_i] A_{i,i} u_i^\top u_i + 2\frac{n(n-1)}{b(b-1)} \sum_{i<j} \mathbb{E}_\xi[\xi_i\xi_j] A_{i,j} u_i^\top u_j.$$

For the first term, we know that $\mathbb{E}_\xi[\xi_i] = b/n$, so

$$\frac{n}{b} \sum_{i=1}^n \mathbb{E}_\xi[\xi_i] A_{i,i} u_i^\top u_i = \sum_{i=1}^n A_{i,i} u_i^\top u_i.$$

For the second term, we first note that

$$\mathbb{E}_\xi[\xi_i\xi_j] = \mathbb{E}_\xi[\xi_i E_\xi[\xi_j|\xi_i]] = \frac{b}{n} \mathbb{E}_\xi[\xi_j|\xi_i = 1] = \frac{b(b-1)}{n(n-1)}.$$

Then we get

$$2\frac{n(n-1)}{b(b-1)} \sum_{i<j} \mathbb{E}_\xi[\xi_i\xi_j] A_{i,j} u_i^\top u_j = 2 \sum_{i<j} A_{i,j} u_i^\top u_j.$$

Adding the two terms back together, we get the trace term.

## B  EXPERIMENTAL DETAILS

All experiments were implemented in PyTorch (Paszke et al., 2019) and PyTorch Lightning, using the normalizing flow implementations provided by Nielsen et al. (2020). In all settings, we use the Adamax optimizer (Kingma & Ba, 2015) and reduce the learning rate with an exponential decay. Weight decay (chosen as described below) is always only applied to the weights of the normalizing flows, not on the dependency parameters $\hat{\lambda}$ and $\hat{\rho}_i$.

### B.1 SYNTHETIC DATA

#### B.1.1 EQUICORRELATED DATA

We sample block-sizes from a Pareto II distribution with shape parameter $\alpha = 0.5$ and minimum value 1, rounded to integer values. We clip block-sizes to a maximum of 1,000 and draw new blocks until all blocks together sum to $n = 10,000$ samples. For each block, we draw one $\rho_i \sim \text{Unif}_{[0.5, 0.99]}$ and define the full covariance matrix as in Section 2.3.2. Using this covariance matrix, we sample non-independently from a bivariate standard normal distribution. We nonlinearly transform these data into complex shapes (`Abs`, `Crescent`, `CrescentCubed`, `Sign`, and `SineWave`) provided by Durkan et al. (2019), for a more challenging density estimation task. We repeat all experiments 10 times with different random seeds.

As a base flow model, we choose rational quadratic spline flows (Durkan et al., 2019), which are state-of-the-art for these challenging data sets. For modelling the equicorrelated blocks, we choose both a grid search over fixed parameters $\hat{\rho} \in \{0.01, 0.025, 0.05, 0.1, 0.175, 0.25, 0.375, 0.5, 0.6, 0.67, 0.75, 0.9\}$ and joint gradient-based optimization of $\hat{\rho}_i$ and the flow parameters, with starting values for $\hat{\rho}_i \in \{0.01, 0.1, 0.25, 0.5\}$. We train all models for 100 epochs, perform a small hyperparameter sweep over learning rate (in $\{0.001, 0.003, 0.01, 0.03\}$) and weight decay (in $\{0.001, 0.01, 0.1\}$), and choose the best model for each setting based on early stopping and validation set performance (which is sampled *without* dependencies).

#### B.1.2 KNOWN COVARIANCE

In this setting, we simulate a known covariance matrix between $n = 5,000$ different samples but with unknown variance component $\lambda$. We first draw a lower-triangular matrix $L$, with diagonals all set to 1 and all elements below the diagonal drawn independently from $\text{Unif}_{[0.5, 0.99]}$. We use $G = \text{norm}(LL^\top)$ as our covariance structure, where norm normalizes the covariance matrix to a correlation matrix (with all-1s on the diagonal). We then use the covariance structure $\lambda I + (1-\lambda)G$ as in the equicorrelated case to generate correlated bivariate standard normal samples that again get non-linearly transformed. We fix $\lambda = 0.5$ in all experiments, and experiments are again repeated 10 times.

For the grid search, we choose $\hat{\lambda} \in \{0.99, 0.975, 0.95, 0.9, 0.825, 0.75, 0.625, 0.5, 0.4, 0.33, 0.25, 0.1\}$ (note that $\lambda$ corresponds to $1 - \rho$) and for the alternating optimization scheme, we initialize $\hat{\lambda}$ from $\{0.99, 0.9, 0.75, 0.5\}$. We train for 100 epochs in the baseline and in grid search; for the alternating optimization, we train for 5 stages of 25 epochs for the flow optimization, with 4 stages of 100 gradient descent updates of $\hat{\lambda}$ inbetween. Remaining parameters are chosen as in the equicorrelated simulations.

### B.2 REAL-WORLD DATA

#### B.2.1 UKB BIOMARKERS

We used an architecture with 16 affine coupling layers, where the fully connected networks have layers `input-128-128-output` for each block, Swish activation functions, and a batch-size of 256, as well as a step-wise exponentially-decaying learning rate schedule.

As in the synthetic experiment, for grid searches we search over $\hat{\lambda} \in \{0.99, 0.975, 0.95, 0.9, 0.825, 0.75, 0.625, 0.5, 0.4, 0.33, 0.25, 0.1\}$ and for the alternating optimization scheme, we initialize $\hat{\lambda}$ from $\{0.99, 0.9, 0.75, 0.5\}$. For both grid search and baseline, we do a hyperparameter sweep over the learning rate (in $\{0.001, 0.003, 0.01, 0.03\}$), weight decay (in $\{0.001, 0.01, 0.1\}$), and number of epochs (25 or 50; in a preliminary exploratory sweep we found that more epochs only lead to overfitting). For the alternating optimization, we chose the same learning rate & weight decay grid, and additionally optimized over the learning rate for $\hat{\lambda}$ (in $\{0.03, 0.1, 0.3\}$) and the number of epochs in the main stages (5 or 25). We alternated for 4 stages, and $\hat{\lambda}$-optimization stages went for 100 steps each.

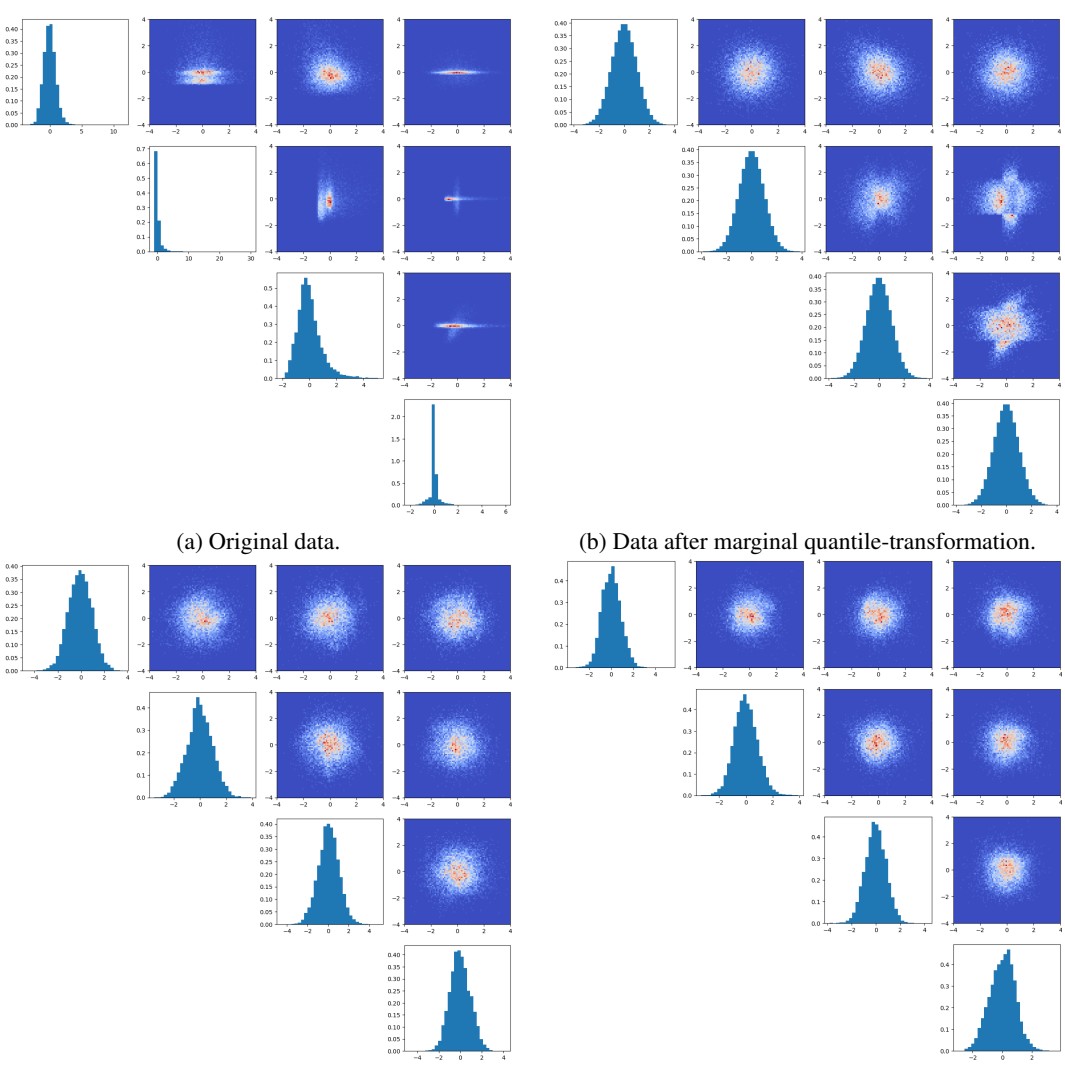

(a) Original data.

(b) Data after marginal quantile-transformation.

(c) Data after transformation with standard normalizing flow.

(d) Data after transformation with normalizing flow correcting for dependencies.

Figure 3: Marginal histograms (diagonals) and 2-d histograms of pairwise joint distributions for the group of "Hormonal" biomarkers.

**GWAS experiment**   For each biomarker group, we selected 10,000 individuals at random from those individuals that had values for the corresponding biomarkers. For flow training, we used the same architecture and training as for the previous Biomarker experiment. Based on the results from the previous experiment, we fixed learning rates at 0.03 (learning rate for $\lambda = 0.1$) and weight-decay on the weights at 0.01. To adjust for fixed covariate effects (age, sex, and genotyping batch), we projected out covariates from the raw phenotypes with a standard linear regression. For the baseline model, we trained for 250 epochs (this performed considerably better than the fewer epochs in the prior experiment). For the alternating flow, we again trained for 4 alternating steps with 25 (flow-stage) and 100 ($\lambda$-stage) epochs each. We performed each experiment three times with random seeds for both the selection of individuals and for flow initialization & data loading.

Figure 3 shows the pairwise joint distributions for the Biomarker group "Hormonal" (after covariates were projected out). Figure 3a shows the original data; Figure 3b shows this data after marginals were transformed using a quantile transformation to standard normal values - it is clearly visible that although the marginals are normally distributed, the joint distribution is far from multivariate normal. Figures 3c & 3d show the data after being transformed with normalizing flows, without and with correcting for dependencies.

Genotype filtering was performed with Plink (Purcell et al., 2007), setting minimum minor allele frequency MAF $\geq 0.1\%$ and Hardy-Weinberg equilibrium p-value $p = 0.001$; and linkage-disequilibrium (LD) pruning with $R^2 = 0.8$ and 500kb window. Both univariate ("Single") and multivariate ("Baseline", "Alternating") GWAS were performed using the GEMMA software version 0.98.5 (Zhou & Stephens, 2012) with score tests (option `-lmm 3`) and centered relatedness matrix (option `-gk 1`). For other GEMMA options we used the defaults, hence, final results were further pruned for MAF $\geq 5\%$. This resulted in approximately 500,000 genotypes per experiment, but slightly varying between different random seeds and different biomarker groups, and a resulting genome-wide significance threshold of $\alpha = 0.05/\text{num\_geno} \approx 10^{-7}$.

Loci were identified using the Plink clumping utility, defining a locus as a group of significantly associated SNPs (single-nucleotide polymorphisms) that were both close spatially (within a 250kb window) and in LD with $R^2 \geq 0.1$.

### B.2.2   Image Modeling

For both data sets we used a Glow-like architecture with 2 scales and 12 steps per scale, as implemented by Nielsen et al. (2020). We grid-searched for $\rho_i \in \{0.01, 0.025, 0.05, 0.075, 0.1, 0.15\}$ and for joint optimization we initialized with the same parameters. ADNI models were trained for 200 and LFW models for 400 epochs, all with a batch-size of 64. All models were trained with a batch-size of 64 and learning rate and weight decay of 0.001 on a single A100 GPU. Due to compute constraints, no further hyperparameter exploration was performed.

**ADNI brain imaging**   The data are T1-weighted MRI, preprocessed and standardized with a brain atlas registration pipeline, using brain extraction, linear alignment, non-linear alignment, and debiasing. The resulting images are more homogeneous than the raw images and thus easier to model. We select the axial-view centered slices and resize them to $64 \times 64$ grayscale images. The $\hat{\rho}_i$ chosen by the best final model with joint optimization ranged between 0.066 and 0.081.

**LFW**   Here, we used $32 \times 32$ RGB images. The $\hat{\rho}_i$ chosen by the best final model with joint optimization ranged between 0.052 and 0.15, while the best model with grid optimization was with $\hat{\rho}_i = 0.15$.

### B.2.3   Stock data pairs

For the stock data, we used an affine coupling normalizing flow with 8 layers of `input-64-64-output` dimensions and swish activation function. Grid search and joint search were initialized with the same values as in the synthetic experiment. We performed a hyperparameter sweep over learning rate ($\{0.001, 0.003, 0.01, 0.03\}$), weight decay ($\{0.001, 0.01, 0.1\}$) and ran all models for 100 epochs and a batch size of 256.

## C  COMPUTATIONAL CONSIDERATIONS

Additional compute & memory requirements for incorporating dependencies depend mostly on the type of dependencies and on the optimization scheme. In our implementation, baseline runs were implemented as special cases of the flow with dependencies (i.e., $\rho_i = 0$ or $\lambda = 1$), which makes fair empirical comparison challenging.

### C.1  EQUICORRELATED BLOCKS

**Grid optimization**  A single run with fixed dependency parameter $\rho_i > 0$ will have almost identical run times as the baseline method with $\rho_i = 0$, as the base distribution likelihood evaluation is not a bottleneck. Since all $\rho_i$ are identical, there is virtually no additional memory requirement. However, as the full network needs to be trained for each of the $M_{\text{grid}}$ grid values tested, the grid evaluation scheme takes roughly $M_{\text{grid}} t_{\text{baseline}}$

**Joint optimization**  In this setting, $N$ (number of blocks) parameters $\rho_i$ need to additionally be estimated and stored in memory, but in all cases considered in this paper this was strongly dominated by the number of parameters in the model (e.g., in LFW, the normalizing flow model had $\sim 90$M parameters, but only a few thousand extra parameters for the individuals. For very slim models and a very large number of blocks, this relationship may change.

### C.2  FIXED COVARIANCE

For the fixed-covariance case, a full spectral decomposition is necessary prior to training, which is (in practice) an $O(n^3)$ operation. It also requires storing the full spectral decomposition in memory. Standard linear algebra libraries used in PyTorch or Numpy & SciPy only support spectral decompositions up to several 10k and oftentimes become unreliable beyond that. Therefore, using fixed covariance schemes is infeasible for larger-scale problems using out-of-the-box software.

**Grid optimization**  For the fixed grid schedule, mini-batch estimation requires quadratic time in the size of the mini-batch, due to the stochastic trace estimator in Equation 2. However, for batch-sizes used in our settings, this was still dominated by the neural architecture shared with the baseline flow architecture. The log-det-Jacobian can be cached and the remaining parts are identical to the baseline flow, so each individual epoch has very similar time requirements to the baseline model. Analogously to the equicorrelated blocks grid optimization, we still need to perform $M_{\text{grid}}$ runs, although the same spectral decomposition can be used for all those runs.

**Alternating optimization**  The main training stage for the flow parameters has identical computational considerations as the grid optimization procedure. However, for optimizing $\hat{\lambda}$ in every other training stage, first the full data set needs to pushed through the normalizing flow and then rotated with the orthogonal matrix $Q^{\top}$ from the spectral decomposition. Despite this, the alternating training procedure was dominated by the original spectral decomposition and the main training stage of the flow.

## D  ADNI IMAGES

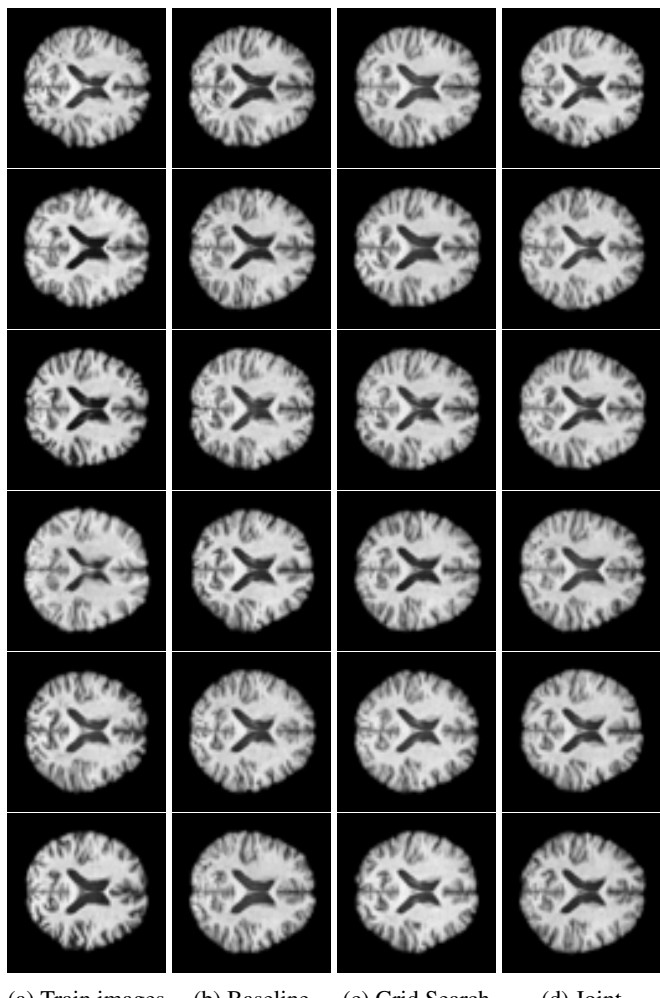

(a) Train images    (b) Baseline    (c) Grid Search    (d) Joint

Figure 4: Random samples of ADNI train images and images generated by the normalizing flow models.

