# OpenReview forum: "Training Normalizing Flows from Dependent Data"
_ICLR.cc/2023/Conference — Submitted to ICLR 2023_

### Official Review · Reviewer_Krj4 · 2022-10-21

**Confidence:** 3
**Correctness:** 3
**Technical Novelty And Significance:** 3
**Empirical Novelty And Significance:** 2
**Recommendation:** 6

**Clarity, Quality, Novelty And Reproducibility:**

- The writing is mostly quite well done except for experiments (see weaknesses)
- Adapting normalizing flows to dependent sampling is novel.
- The authors provide code that will make it reproducible [I have not tried running their code.]


**Strength And Weaknesses:**

Strengths:

- The authors chose an oft ignored area in generative modeling: that of dependent sampling which is interesting to analyze.
- The paper is mostly well written and flows well.

Weaknesses:
- I am not quite sure why the specific covariance structures were chosen. As an aside do the authors think that adding a regularisation functional that minimises the norm of $\rho$ will benefit the joint optimization strategy. In a way, would that choose the most decorrelating bijective map?
- The authors should provide a compute budget difference (both time and memory) between the baseline and proposed methods. It seems the baseline is just a quadratic spline flow whereas the proposed method has additional parameters for the latent correlation structure. An aside: have the authors attempted to optimize the baseline latent mean and covariance?
- The experiment section lacks many details:
  -- In UKB: how is the relatedness used? Do you assume G is known?
  -- In stocks: a few samples would help  understand what is log return referring to? Is it tracking per day movement of a pair? How is the temporal nature of stocks over time incorporated in the model?
  -- ADNI: To my eyes all images in figure 3 look very similar. Does dependent modeling help in downstream tasks? If so, it would strengthen the author’s case a lot if they could show something of that sort.

Overall I feel the authors would greatly benefit in relegating some exp details to the appendix (around values in grid search, extra dataset details) and adding more results from each of the datasets. For ex. in UKB they say a quantile transformation makes statistical modeling hard. It would strongly benefit the authors to show that their way of modeling makes that easier.



**Summary Of The Paper:**

The paper tries to tackle the problem of learning a generative model for sampling identically distributed but dependent data points from a distribution. Like previous normalizing flows, they try to learn a bijective map of data to an assumed latent space but unlike previous works which assume a diagonal Gaussian, they assume a correlation structure within the latent space as well. This makes the computation of probability density estimates in the latent space intractable in general but by restricting to specific classes of correlation structures the authors design a computation scheme that works for their case of dependent data. The authors test their method on toy and real world datasets.


**Summary Of The Review:**

I feel the authors need to strengthen the experiments section considerably in an attempt to justify that dependency modelling is actually useful [other than slight NLL benefits]. Other than that I think the paper is a good attempt at incorporating data dependency assumptions into the normalizing flow literature. I am willing to change my rating based on discussions on the forum.

---

> ### Author Response · Authors · 2022-11-18
> **Response to Reviewer Krj4**
>
> > "I am not quite sure why the specific covariance structures were chosen."
>
> - The structure $\lambda I + (1-\lambda) G$  with fixed $G$ was chosen because it is highly flexible (there are no restrictions on $G$ except that it is positive semidefinite), and in many applications, we *do* have explicit knowledge of the *relative* pairwise dependencies. For example, in genetics, $G_{i,j}$ may be the correlation between the individual $i$'s and $j$'s measured genotypes. However, we may not know how large the overall effect on the trait is, so we have to fit the variance component $\lambda$. This structure is also computationally convenient: it requires a one-time spectral decomposition beforehand, but after that, fitting this model incurs only a little overhead. In some settings, we might have two relationship matrices, $G_1$ and $G_2$ with $C = (1-\mu_1 - \mu_2) I + \mu_1 G_1 + \mu_2 G_2$ (e.g., one for general population stratification, the other for pedigree-derived relatedness). In this case, estimation of $\mu_1, \mu_2$ can be considerably more involved, as the trick with the spectral decomposition does not apply. Developing optimization algorithms for this more general case may be an interesting avenue for future work.
> - The equicorrelated block structure, on the other hand, is extremely efficient and incurs almost no additional computational costs. It is also a good model for repeat measurements, which is very common in practice and a standard application of random effects/dependency modeling in classical statistics.
>
> Other reasonable dependency structures may be, e.g., band matrices for time series or certain non-diagonal block structures.
>
> > "do the authors think that adding a regularisation [...]"
>
> Regularization of the free parameters $\rho$, e.g., with an $L_{p\geq 1}$ norm, is certainly possible. This would constitute an inductive bias towards smaller and potentially sparse correlation structures, which may or may not make sense for a specific setting.
>
> > "[...] compute budget difference [...]"
>
> Additional compute & memory requirements for incorporating dependencies depend mostly on the type of dependencies and the optimization scheme. In our implementation, baseline runs were implemented as special cases of the flow with dependencies (i.e., $\rho_i = 0$ or $\lambda = 1$), which makes fair empirical comparison challenging. In all cases, the biggest computational impact was due to the grid searches (i.e., we needed to run the same model multiple times with different $\rho$/$\lambda$ values) and due to the spectral decomposition necessary for the fixed-covariance scheme. We added more details to the appendix.
>
> > "[...] optimize the baseline latent mean and covariance?"
>
> We did not optimize over the marginal mean & covariance structure. To the best of our knowledge, for normalizing flows, this is done very rarely directly due to added computational complexity. Instead, the standard transformations in the normalizing flow (such as affine coupling layers) are generally assumed to be flexible enough to model arbitrary distributions with high precision. They should certainly be able to (approximately) map from the more general $\mathcal{N}(\mu, \Sigma)$ to a simpler $\mathcal{N}(0, I)$.
>
> > "The experiment section lacks many details [...]"
>
> Thank you for pointing out these problems. We added all the missing details in Section 3 and the appendix. Here are the answers in short:
> - UKB: yes, $G$ is assumed known and computed from the correlation of the first 40 genetic principal components (which captures population stratification very well, but direct relatedness less so).
> - Stocks: Yes, it is the logarithmic change of price from one day to the next, at close. We did not take into account temporal dependencies directly.
> - Training images were preprocessed with a pipeline that registers those images to a standardized brain atlas, a common procedure on MRI. The resulting images are relatively homogeneous, and differences are harder to discern for non-experts. However, the main advantage of our method is not that "individual images look better" but rather that "the full distribution is better represented," as measured by the full test-set likelihood. As a typical example, GANs may produce realistic images (i.e., each individual image looks good) but can exhibit considerable mode collapse (i.e., the full distribution is not well-represented).
>
> > "Does dependent modeling help in [...]"
> > "Overall I feel the authors would  [...]"
>
> Thank you for these suggestions. We now moved implementation details to the appendix, added a new experiment that shows that our model considerably boosts downstream performance in an important association testing task, and added additional clarifications to the other experiments. Please see the main response to all reviewers and the updated manuscript for details.

---

### Official Review · Reviewer_xwvU · 2022-10-25

**Confidence:** 4
**Correctness:** 4
**Technical Novelty And Significance:** 3
**Empirical Novelty And Significance:** 3
**Recommendation:** 6

**Clarity, Quality, Novelty And Reproducibility:**

The paper is well written. However, it could help make the paper stronger to focus and commit on one application problem more deeply, use it as a clear consistent running example, show strong results on it, etc. The results seem to be novel, high quality, and reproducible.

**Strength And Weaknesses:**

STRENGTHS:
* The paper studies an important problem. Although the motivation describes it as non-IID data, this method also applies to high-dimensional correlated vectors, and may be useful for modeling IID data with interesting correlations among the features.
* This work is novel, and to my knowledge this techniques of extending copula to higher-dimensional datasets is new.
* The algorithmic ideas proposed in the work are novel, technically sound, and non-trivial. The level of technical depth required to derive the methods is significant.
* The experiments are extensive and look at a wide range of interesting real-world problems.

WEAKNESSES:
* These are all relatively minor comments. I think it would have been interesting to describe the motivation and the various applications of the method early on (in the introduction). I only "got" the idea and why it's interesting after reading the whole paper. I didn't feel like the intro did a good job at selling the work.
* While the experiments look at many domains, I didn't feel like the method achieved very strong improvements. Highlighting specific examples where this method really shines could make the paper stronger.
* The paper mentions GWAS and ancestry-based confounding a lot, and it sounds like an interesting motivating application. It could help make the paper stronger to focus and commit on one application problem more deeply, use it as a clear consistent running example, show strong results on it, etc.


**Summary Of The Paper:**

The authors propose an extension of normalizing flows to non-IID data. This extension can be seen as the generalization of a Gaussian copula to vector-valued variables. The authors propose parameterizations of the resulting model and associated learning algorithms that yield efficient training.

**Summary Of The Review:**

I can see this paper being a useful addition to this line of literature, although I don't see it as completely ground-breaking, It could benefit from a more developed and more thoroughly studied use case.

---

> ### Author Response · Authors · 2022-11-18
> **Response to Reviewer xwvU**
>
> Thank you for the kind words and the suggestions for improvements that helped us improve upon the paper. We greatly expanded on the UKB Biomarker application and introduced a new downstream experiment, performing GWAS between biomarker groups and genetic variants. In this application, our proposed method yielded >8% more association results than the baseline methods. Please see the primary response to all reviewers and the updated manuscript for details. We also adapted the introduction and hope the motivation and applications are more apparent now.

---

> > ### Comment · Reviewer_xwvU · 2022-12-08
> > **Thank you**
> >
> > I acknowledge the response and am increasing my score by +1

---

### Official Review · Reviewer_gW39 · 2022-10-26

**Confidence:** 4
**Correctness:** 4
**Technical Novelty And Significance:** 3
**Empirical Novelty And Significance:** 3
**Recommendation:** 3

**Clarity, Quality, Novelty And Reproducibility:**

The paper is clear in terms of the proposed approach. The specific tasks being solved require more motivation. To my knowledge, introducing dependency to normalizing flows novel, but I am not an expert in the normalizing flows literature. Many of the technical results are "textbook" decompositions of covariance matrices that are well-trod in the ridge regression and GP literatures (probably among others). The experiments seem easily reproducible.

**Strength And Weaknesses:**

Strengths:
 + Dependence between datapoints can be an important issue, especially in cases where one cares about being able to make joint or conditional predictions where the task depends on multiple datatpoints simultaneously.
 + The derivations are clearly explained. Although they are straightforward, they are not obfuscated to appear more complicated than they are.
 + Several practical issues in training are addressed.
 + Experiments explore degrees of freedom that were raised in the development of the method.

Weaknesses:
  - The primary weakness to me is that the method seems undermotivated. Although I know that there exist tasks where dependency between datapoints matter, it is not clear how accounting for dependencies between data translates to better solutions for important tasks. The authors raise several applications, including genomics, finance, and neuroimaging, but stop short of indicating what problems could be solved in these areas if only we modeled dependencies between examples better. NLL is not itself self-motivating; it measures the model's consistency with the data, but does not tell us what this extra fit buys in terms of performance on a real task. What is the unsolved problem in genomics, finance, or neuroimaging that can be solved with a good model of between-example dependence?
 - Along the same lines, when we consider dependent data, the actual prediction task needs to be specified with more care. What exactly is being measured by performance on the test data? Do we care about our having low expected prediction risk on the next datapoint? Low joint prediction risk on a set of $m$ new datapoints? Some kind of statistical inference? In the first case, dependency structure would not matter. In the second case, it might matter, but one would need to motivate why we care about $m$ datapoints simultaneously, and the composition of this test set would need to be motivated carefully (e.g., would this test set represent measurements from a completely new individual? from an individual we have observed before?). Also, we might need the test set to be composed of $n$ replicates of $m$ datapoints. In the third case, the form of statistical inference would need to be made clear (e.g., Are we doing anomaly prediction and constructing p-values? Are we using the normalizing flow to do covariance estimation for a downstream regression / GWAS task?).
 - For example, it is not clear what the goal of the task illustrated in Figure 1 is. If the goal were density estimation, it seems like the standard normalizing flow is doing the right thing. Meanwhile, it is not clear why we want the dependency-adjusted normalizing flow to match the distribution of data sampled independently. This comes back to the issue that when one has dependent data, the goals are not self-evident in the same way that they are when predicting with independent data. Goals and notions of bias need to be made explicit.
 - Given the note in the discussion that "equal block sizes" would be a case where the dependent data objective would produce no changes, did the authors consider the baseline of weighting datapoints by inverse block size? It seems that for some tasks this might address many issues and not require major modifications to training.
 - Many tricks for working with covariance matrices have been discussed before in literatures that do efficient ridge regression or work with Gaussian processes. I am not sure that the linear algebra tricks need to be rehashed in the main body of the paper, and this could make room for better discussion of tasks where dependence matters.

**Summary Of The Paper:**

The paper proposes a modification to the standard normalizing flow training loss to account for certain dependencies in data. Specifically, the authors consider the special case where datapoints are dependent in latent space, but are transformed pointwise to the observed space. In addition, the authors specialize to the case where the correlation structure is either known up to a convex combination with the identity, or has a block-diagonal structure with known block assignment and constant positive correlation within blocks. The authors then propose training strategies for learning the parameters of these dependencies. The authors evaluate experimentally on simulated and real datasets, using NLL on held-out datasets as the core metric.

**Summary Of The Review:**

The proposed method seems fine in terms of specifying an objective that reflects certain forms of independence. However, the actual tasks being solved require far more explicit discussion, since there are many different tasks that one might try to solve in a dependent data context, and it is not clear which of these the method is designed to address. Because of this concern, the paper feels incomplete.

---

> ### Author Response · Authors · 2022-11-18
> **Response to Reviewer gW39**
>
> Thank you for these thoughtful comments. We addressed most of these points in our response to all reviewers; specifically, we introduced a new downstream experiment for the biomarker use case, which showed a considerable improvement; we added more context on why better likelihood fits on the stock market data is important; and we discussed sampling bias in imaging models and also included another imaging experiment.
>
> Regarding your query on prediction tasks: we especially consider models whose main use is not necessarily on *prediction tasks*, but rather in *statistical inference* tasks for the biomarker & stock modeling examples and *generative modeling* in the image examples. So far, we didn't look into joint prediction risk on downstream tasks, but we agree that this may be an interesting future research direction. Naturally, as you mentioned, there may be other tasks where correcting for dependencies is undesirable.
>
>
> > "Given the note in the discussion that "equal block sizes" would be a case [...]"
>
> For special cases of repeat measurements, this may work. For other settings with more general covariance structures, or even with equicorrelated blocks that exhibit different correlation strengths in different blocks (independent of block size), this scheme may break down, though. Our proposed solution is more general by allowing *arbitrary* correlation structures. We only considered the two cases of equicorrelated blocks and convex combination of fixed covariance and identity, as those two frequently appear in applications. But many other covariance structures can be directly incorporated into our model naturally and only require adjustment to the base NLL computations. This is not true for a specialized ad-hoc solution such as a simple reweighting.
>
> > "Many tricks for working with covariance matrices have been discussed [...]"
>
> Thank you for the suggestion; we tightened Section 2 and instead included more detailed discussions in Section 3.

---

### Author Response · Authors · 2022-11-18
**Main Response**

We are very grateful for the valuable feedback all reviewers provided. All three reviewers shared a very similar main critique: namely that it is not entirely clear what the actual benefit of our proposed method is, apart from a better likelihood fit. In response, we have made several changes to the manuscript that better point out the relevance of adjusting for data dependencies. We introduced new experiments, rewrote certain passages, and included new explanations in this revision.
We focus on one new downstream application of the biomarker use case and show that our proposed method increases the statistical power of a genome-wide association study (GWAS) by >8%, a considerable improvement. Additionally, for the image modeling use case, we included one more data set for which we believe the dependency adjustment is more intuitive than in the brain MRI data set. For the stock market model, we also provide additional explanations both here and in the manuscript that we hope will make the relevance of our method clearer.
We summarize our changes to the manuscript in this response and answer reviewer-specific questions in the individual responses.

## UKB Biomarker downstream analysis
In Section 3.2.1, we added a new downstream analysis. Namely, we perform GWAS on subgroups of the biomarkers after either quantile transformations or gaussianization with normalizing flows. Due to the unique properties of normalizing flows, we can transform arbitrarily distributed data into multivariate normally distributed data. This allows for applying multivariate linear mixed models in genome-wide association studies to these data. By contrast, data transformed classically with univariate quantile transformations will not be multivariate normally distributed and does not satisfy the statistical requirements for multivariate analysis. We apply both a standard normalizing flow, as well as our normalizing flow with dependency adjustments to this task. We find that adjusting for the data dependencies already in the flow, and not only in the linear mixed model, boosts the power of the GWAS by 8-9%, both against the normalizing flow baseline and against the traditional quantile transformation with univariate GWAS.

## Generative Imaging
Image modeling is one of the larger application areas of normalizing flows in the current literature.
Many image data sets exhibit considerable sampling biases. We showed the example of the ADNI brain imaging study, where some individuals were recorded more often than others. For a more intuitive example, consider the setting of facial image synthesis, such as with the Labeled Faces in the Wild (LFW) data set. LFW consists of face images from celebrities, but some individuals are grossly overrepresented. For example, George W. Bush has 530 images, while around 70% of the people in the database only appear once. A density model trained without adjustment for dependencies would put much more probability mass on images similar to George W. Bush, compared to those individuals that are only once in the data set.
Two prominent use cases for normalizing flows on images are *image generation* and *outlier detection*. We would argue that in both settings, practitioners would often prefer a model that is less biased towards overrepresented individuals. A similar argumentation holds when considering the more general dependencies, such as for medical images of individuals that are genetically related.

## Stock market modeling
Accurate probabilistic modeling of the joint distribution of the returns of individual stocks is extremely important in financial engineering. In the use case studied in our manuscript, we consider the daily returns of two pairs of correlated stocks (`AAPL-MSFT` and `MA-V`). A pair trading strategy would assess at each time step if the pairs currently trade outside of a high-confidence region, based on the probabilistic model. If the pair indeed behaves anomalously, one can often hedge one against the other (going "long" the one and "short" the other) until the pair trades again in a "normal" range. In the example, there is abundant training data available for `AAPL-MSFT`, but much less for `MA-V`, as those stocks were issued much later to the public. However, once the model is deployed, both pairs are equally common. A model trained without adjusting for dependencies will overweigh one pair over the other and give biased results on the test set.
Reliable backtesting of trading strategies requires a lot of careful engineering and is, unfortunately, beyond the scope of this submission. However, we added more explanation of this context to section 3.2.3.

## Other changes
- more context in the introduction
- for a more careful discussion of the case studies, we tightened Section 2 and moved parts of section 3 to the appendix
- more details on the experimental setups in the appendix
- a new section on computational considerations in the appendix.

---

### Decision · Program_Chairs · 2023-01-20

**Decision:**

Reject

**Justification For Why Not Higher Score:**

Despite being novel the reviewers do not sufficiently appreciated the paper. This is a bit harsh but there is another conference deadline in a month's time so not the end of the world.

**Justification For Why Not Lower Score:**

N/A

**Metareview: Summary, Strengths And Weaknesses:**

The paper proposes to model data as identically but not independently distributed using a flow model. This means that the prior over the latent variables are dependent but the mapping from latent to data is ii. The Jacobian then still factorises but the prior distribution has dependencies.

This is a neat idea. However, the reviewers have several issues with the paper: motivation - why dependence, no convincing results over iid baselines, too unfocused range of experiments.

Overall, the arguments against the paper from the reviewers are not very strong but they reflect a general trend that indicates that the authors could have communicated their contributions better.